# Geometry Cloak: Preventing TGS-based 3D Reconstruction from Copyrighted Images

**Qi Song**[1,2], **Ziyuan Luo**[1,2], **Ka Chun Cheung**[2],
**Simon See**[2], **Renjie Wan**[1]*

[1]Department of Computer Science, Hong Kong Baptist University
[2]NVIDIA AI Technology Center, NVIDIA
`{qisong,ziyuanluo}@life.hkbu.edu.hk`
`{chcheung,ssee}@nvidia.com, renjiewan@hkbu.edu.hk`

## Abstract

Single-view 3D reconstruction methods like Triplane Gaussian Splatting (TGS) have enabled high-quality 3D model generation from just a single image input within seconds. However, this capability raises concerns about potential misuse, where malicious users could exploit TGS to create unauthorized 3D models from copyrighted images. To prevent such infringement, we propose a novel image protection approach that embeds invisible geometry perturbations, termed "geometry cloaks", into images before supplying them to TGS. These carefully crafted perturbations encode a customized message that is revealed when TGS attempts 3D reconstructions of the cloaked image. Unlike conventional adversarial attacks that simply degrade output quality, our method forces TGS to fail the 3D reconstruction in a specific way - by generating an identifiable customized pattern that acts as a watermark. This watermark allows copyright holders to assert ownership over any attempted 3D reconstructions made from their protected images. Extensive experiments have verified the effectiveness of our geometry cloak. Our project is available at `https://qsong2001.github.io/geometry_cloak`.

## 1 Introduction

With the increasing importance of 3D assets, several methods have been proposed to reconstruct or generate 3D models from single 2D images. Combining with Tensorial Radiance Fields [2], Triplane-based Gaussian Splatting (TGS) [52] presents a compelling approach for producing 3D models from single-view images. However, malicious users could potentially exploit TGS [52] to generate 3D models from single-view images without authorization, posing a threat to the interests of image copyright owners. To address this issue, it is essential for image owners to implement measures that can safeguard their copyrighted images from being used by TGS [52].

Digital watermarking [6, 51] is an effective way of claiming the copyright of digital assets. Thus, one potential method for safeguarding copyrighted images is to embed unique messages within images intended for building a 3D model and then extract the embedded messages from the reconstructed 3D model. However, previous methods have proven it is difficult to transfer the embedded copyright messages in 2D images into 3D models [31, 19]. Moreover, even if we can embed and extract copyright messages, the 3D models might have already been used by others before the copyright being claimed.

To prevent unauthorized 3D reconstruction from copyrighted images via TGS [52], a possible approach is to incorporate adversarial perturbations [11, 28] into input images intended for TGS.

---

*Corresponding author.

38th Conference on Neural Information Processing Systems (NeurIPS 2024).

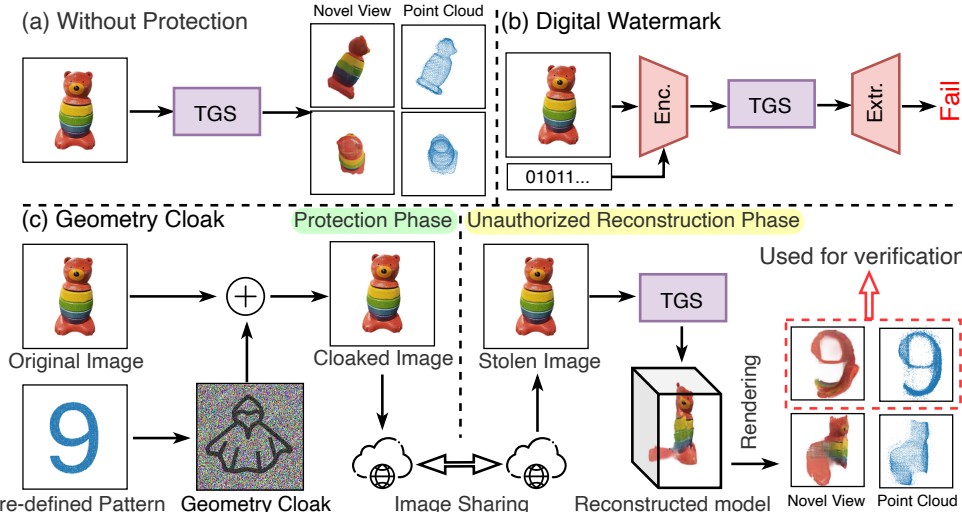

Figure 1: Overview of our scenario. (a) Images without protection. Images can be easily reconstructed into 3D models by malicious users with TGS [52], posing a threat to the copyright of the image owner. (b) Digital Watermarking offers a solution by embedding copyright messages into the view-image before 3D reconstruction. However, the embedded message cannot be extracted from novel rendered views. (c) Geometry Cloak. Our geometry cloak utilizes the disturbance-prone components of TGS, achieving view-specific watermark embedding. Our method can compromise the unauthorized reconstructed 3D model while providing a verifiable pattern for copyright claim.

Adversarial methods [11, 28] have already achieved promising results by introducing disturbances into input images to prevent neural models from functioning correctly. When it comes to TGS [52], a straightforward solution is to incorporate such adversarial perturbations into input images by maximizing the difference between rendered and ground truth views like previous methods [9, 13]. However, those methods [9, 13] intuitively focus on disturbing rendered results, ignoring perturbation-prone components of TGS [52]. As a result, those conventional adversarial perturbations can only lead to limited changes to the reconstructed results [9, 13], which may still be used for several illicit applications. Besides, simple adversarial perturbations can only disrupt the results of 3D reconstruction in an uncontrollable manner but do not support traceability. The users have difficulties claiming copyright after the disturbance-affected 3D generation of images.

We envision a novel scenario where copyrighted images can induce TGS to generate compromised content with an identifiable pattern. To achieve this goal, a naive solution is to employ image cloaking. Traditionally, image cloaking techniques [23, 37, 41, 35] are used to prevent the image from malicious editing by diffusion models [33]. The cloaking integrated into images can shift image features into another domain through adversarial perturbations. Consequently, this manipulation directs diffusion models to generate predetermined specific outcomes. If malicious users attempt to create 3D models using these protected images, the resulting compromised 3D models will be unusable. Besides, the identifiable pattern exhibited can help the image owner assert their copyrights in the event of legal inquiries.

However, unlike image cloak methods [35, 23] against diffusion models [33], simply perturbing image features cannot effectively induce TGS (as shown in Fig. 1). Image features in TGS show strong robustness against disturbances, even under relatively strong attack settings. Thus, the key becomes how to identify perturbation-prone components in TGS [52] and capitalize on this weakness to induce reconstructed results. TGS [52] contains image and geometry features during its 3D model reconstructions. Previous works [8, 3] have shown the geometry feature in 3D Gaussian Splatting [18] is easier to be manipulated with external operations. Therefore, we wonder if it is possible to manipulate the estimated point cloud process with an invisible disturbance. Based on this simple observation, we propose embedding invisible adversarial perturbations as a geometry cloak on images intended for TGS [52], which can affect and manipulate estimated point clouds in the process of TGS's 3D reconstruction.

As shown in Fig. 2, we introduce a geometry cloak, which is carefully crafted to induce TGS [52] to fail 3D model generation and reveal our embedded pattern. To induce the TGS to reveal the embedded patterns, we propose a view-specific Projected Gradient Descent [28] (view-specific PGD) strategy by optimizing the distance between the projected point cloud view and predefined patterns. The PGD iteratively updates the geometry cloak by minimizing CD loss, ultimately revealing the desired view and uncovering the hidden patterns within the TGS [52]. When malicious users reconstruct 3D models using the protected images with TGS, the compromised geometry information causes TGS to reveal the embedded message. Unlike traditional copyright protection methods like digital watermarking [25, 51, 39], which typically entail additional procedures for extracting the watermark post-use. Our approach directly transforms TGS [52] into a message disclosure tool by inducing it to yield a specific stylistic outcome, facilitating the verification of image ownership rights. In summary, our main contributions are threefold:

- We propose the concept of "geometry cloaking", which can prevent unauthorized image-to-3D generation by TGS [52], and it also leaves a verifiable copyright pattern.

- Our geometry cloaking technique explores the perturbation-prone components of Triplane-based Gaussian Splatting and utilizes this vulnerability to achieve the protection of images.

- We propose a view-specific PGD strategy, which can embed identifiable patterns into a specific view of the reconstructed 3D model.

As our approach attacks the geometry features that are widely used in various single-image-to-3D approaches, it demonstrates generalization capability to other GS-based single-view to 3D approaches like LGM [40]. The results can be found in our experiments.

## 2 Related work

### 2.1 Building 3D models from single images

There has been a surge of research on learning to generate novel views from a single image [47, 36, 52]. This task aims to infer the 3D structure of a scene from a single 2D image and render new perspectives, enabling applications in virtual reality, augmented reality, and computer-aided design. One approach is NeuralLift-360 [47], which incorporates a CLIP loss [32] to enforce similarity between the rendered image and the input image. Another method, 3DFuse [36], fine-tunes the Stable Diffusion model [33] with LoRA [14] layers and a sparse depth injector. Recent work like Zero123 [24] takes a different approach by fine-tuning the latent stable diffusion model [33] to generate novel views based on relative camera pose. Recently, Triplane-based neural rendering methods [2, 52, 12] adopt a novel approach to model and reconstruct radiance fields. Unlike NeRF, which uses pure MLPs, Tensorial Radiance Fields (TensoRF) [2] consider the full volume field as a 4D tensor and propose to factorize the tensor into multiple compact low-rank tensor components for efficient scene modeling. Combing TensoRF [2] with novel 3D Gaussian Splatting [18], Zou *et al.*propose a Triplane-base Gaussian Splatting [52], which can obtain a 3D model from single-view image within seconds [52]. With continued research and development, more impressive and realistic 3D reconstructions in the future will enable immersive experiences and streamlined design workflows. The advancements in creating 3D models from a single-view image offer significant potential for diverse applications and digital assets. Consequently, it is essential to address the protection of copyrighted images to prevent their misuse in generating 3D models.

### 2.2 Adversarial attacks for neural rendering

Adversarial attacks [11, 28] have become a significant concern in the field of computer vision [22, 16]. These attacks aim to deceive machine learning models by introducing adversarial perturbations to the input data, leading to incorrect predictions such as misclassification [11]. While initially studied in the context of image classification models, adversarial attacks have also been explored in other domains [35, 23]. For Neural Radiance Fields (NeRFs) [29], several works have proposed methods to perturb or enhance the original NeRF framework. NeRFs are a class of models that can synthesize high-quality 3D scenes from 2D images by learning the scene's volumetric representation and appearance. However, NeRFs are also vulnerable to adversarial attacks. Several recent works have investigated different techniques to perturb or enhance the original NeRF framework using adversarial

attacks. NeRFool [9] presents an approach to manipulate NeRFs by perturbing the scene's geometry and appearance using adversarial attacks. NeRFTargeted [13] focuses on targeted perturbations in NeRFs, allowing for the optimization of scene parameters to generate desired target images. ShieldingNeRF [46] introduces a technique to protect sensitive information in NeRF-generated views by introducing obfuscating perturbations. These works demonstrate the potential of perturbing NeRFs for various objectives, including adversarial attacks, targeted image manipulation, and privacy protection, contributing to the advancement of NeRF-based models [2, 29, 27] in computer graphics and computer vision tasks. However, with the promising developments in 3DGS [18], research on defending against attacks on Gaussian Splatting renderings remains an area that requires further investigation.

### 2.3 Image cloaking

With the rapid advancement of AI models, the risk of misuse has raised concerns, particularly in the malicious use of public data [43, 42, 50, 17]. Researchers advocate a proactive approach to prevent such misuse by adding subtle noise to images prior to publication. This technique aims to disrupt attempts at exploitation [21, 26, 20]. An important application of image cloaking is to prevent privacy violations from face recognition systems [38, 4, 15]. By introducing noise patterns to facial regions, these methods degrade face recognition model performance while maintaining visual quality. Image cloaking can also thwart image manipulation through GAN-based techniques like DeepFakes [30] by corrupting latent representations. Recent image cloaking methods [37, 23, 41] focus on protecting the copyrighted image from misuse by stable diffusion models. These methods disrupt artistic mimicry and harmful personalized images, aiming to prevent unauthorized exploitation. For example, GLAZE [37] and AdvDM [23] primarily focus on disrupting artistic mimicry and harmful personalized images generated by text-to-image models[10]. Anti-Dreambooth [41] concentrates on fine-tuning DreamBooth [34] for malicious face editing. These techniques aim to cloak input images in a way that disrupts the model's ability to generate personalized content while preserving the overall visual quality. Existing methods focus on protecting the copyrighted image from misuse by disturbing image features. By contrast, our method focuses on preventing copyrighted from being 3D reconstructed by TGS [52] without authorization, facing 3D scenes and complex copyright validity verification settings.

## 3 Preliminaries of TGS

TGS [52] introduces a novel hybrid 3D representation that integrates an explicit point cloud with an implicit triplane, enabling efficient and high-quality 3D object reconstruction from single-view images. As shown in Fig. 2, the representation consists of a point cloud $\mathcal{P} \in \mathbb{R}^{N \times 3}$ providing explicit geometry, and a triplane $T \in \mathbb{R}^{3 \times C \times H \times W}$ encoding an implicit feature field, where $T = (T_{xy}, T_{xz}, T_{yz})$ comprises three orthogonal feature planes. For a given position $x \in \mathbb{R}^3$ from the point cloud, the corresponding triplane feature $f_t$ is obtained by trilinear interpolation and concatenation of features from the three planes:

$$f_t = \text{interp}(T_{xy}, \mathcal{P}_{xy}) \oplus \text{interp}(T_{xz}, \mathcal{P}_{xz}) \oplus \text{interp}(T_{yz}, \mathcal{P}_{yz}). \tag{1}$$

Utilizing this hybrid representation, the 3D Gaussian attributes like opacity $\alpha$, anisotropic covariance (scale $s$ and rotation $q$), and spherical harmonics coefficients $sh$ are decoded from $f_t$ augmented with projected local image features $f_l$ using an MLP $\phi_g$:

$$(\Delta x', \alpha, s, q, sh) = \phi_g(x, f_t \oplus f_l). \tag{2}$$

Together, these parameters parameterize the 3D Gaussian kernel attributes around the point $x$, enabling differentiable Gaussian splatting for rendering.

TGS represents a cutting-edge approach to 3D object reconstruction from single-view images, combining explicit and implicit representations to achieve accurate and detailed reconstructions. By leveraging advanced decoding mechanisms and efficient rendering techniques, the model demonstrates superior performance in generating realistic 3D models with intricate geometry and textural details.

## 4 Methodology

**Overview.** With the increasing capabilities of 3D reconstruction techniques like TGS [52], there is a risk of malicious users exploiting these methods to generate 3D models from copyrighted images

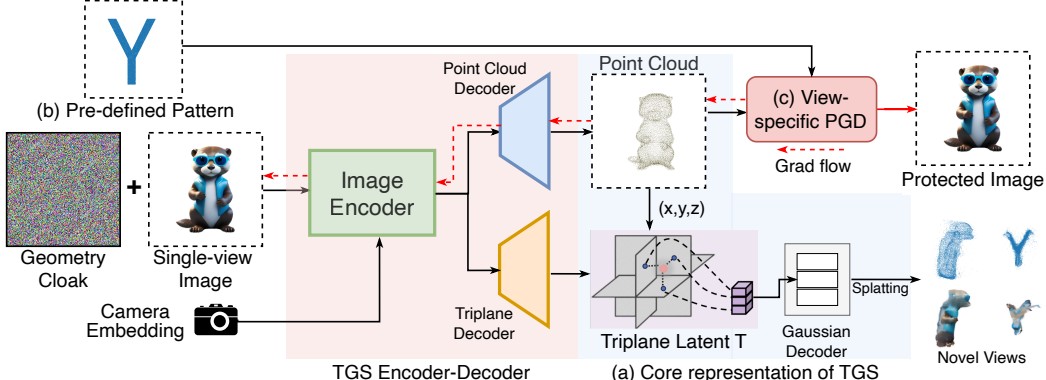

Figure 2: Overall of our proposed method. We propose to induce the 3D reconstruction process with our geometry cloak. (a) The core representation of TGS [52] includes an explicit point cloud and an implicit triplane-based feature field. The features of the novel view image are extracted through the coordinates in the point cloud. (b) The target patterns (Section 4.1) are designed to induce the final reconstruction result. (c) In order to make the reconstruction result show some distinguishable characteristics, we use projected gradient descent (PGD) [28] to iteratively optimize the reconstructed point cloud so that it has consistent characteristics with the target point cloud (Section 4.2).

without authorization, infringing on the rights of image owners. We propose geometry cloak, a novel solution by embedding invisible perturbations on the input images intended for TGS [52]. These perturbations are crafted to induce TGS to fail the reconstruction in a distinct way, producing an identifiable pattern in the corrupted 3D output.

This section presents the methodology of inhibiting 3D reconstruction of TGS [52] via the introduced geometry cloak. Our method consists of two stages: (1) Building verifiable geometry patterns (Section 4.1), and (2) Optimizing geometry cloak with view-specific PGD (Section 4.2).

## 4.1 Building verifiable geometry pattern

As illustrated in Fig. 2, to obtain a 3D model from single-view images, TGS [52] encodes the single-view image $\mathcal{I}$ and its associated camera parameters into image features. Following this, a point cloud decoder is adopted to project image features onto the point cloud. Then, a triplane decoder converts the image feature into the triplane latent $T$. Finally, 3D Gaussians are decoded from triplane feature $f_t$ for novel view synthesis.

Our methodology diverges by targeting the explicit geometry features of the point cloud. Perturbing the point cloud directly is inherently more effective due to the inherent vulnerabilities in the TGS reconstruction process. In TGS [52], the point cloud offers a distilled and direct representation [18] of the scene's structure, making modifications on them quite evident [8, 3]. Besides, the point cloud is not only foundational geometry information but also typically helps subsequent processing to obtain the final output. Point clouds provide essential geometry information that is instrumental in the sampling of features from the latent triplane representation (Eq. (1)). Moreover, by introducing strategic alterations to the geometry feature, we bypass some of the inherent robustness found in image-level features (Table 1) and avoid the complex transformations between image and geometry spaces. This direct manipulation allows for more precise control over the adversarial impact, exploiting specific vulnerabilities in TGS and leading to more pronounced disruptions in its output.

**Different types of target geometry pattern.** Based on the inherent vulnerabilities of point clouds, we propose two types of verifiable pattern (Fig. 3), including (1) *Pre-defined patterns* and (2) *Customized patterns*. Through these patterns, the reconstructed 3D model undergoes consistent changes in geometry and visual appearance (novel views).

*Pre-defined patterns* are 2D point clouds that are transformed from alphanumeric characters images, creating a direct and straightforward representation of watermarks. To obtain these 2D point clouds, we segment the image of alphanumeric characters and record the coordinates of these alphanumeric

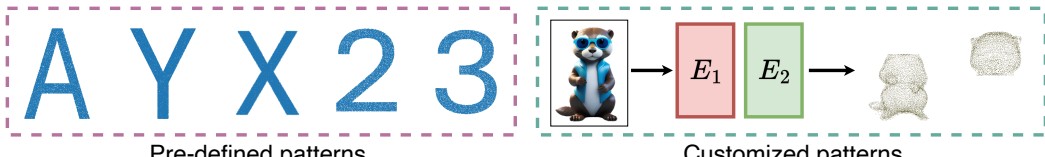

Pre-defined patterns                        Customized patterns

Figure 3: Two different target geometry patterns. (1) Pre-defined patterns: we directly convert alphanumeric characters into a 2D point cloud as watermarks. (2) Customized patterns: In $E_1$, we first extract the point cloud of the image that needs to be protected. In $E_2$, we edit the acquired point cloud through text-guided methods like instructP2P [48] or open-source software meshlab [5].

characters' segmentation. Then, we sample the point from segmentation coordinates and form a 2D point cloud as pre-defined patterns.

*Customized patterns* are a more personalized approach where users can selectively protect a certain part of an image. In this setting, we first extract a point cloud from the image that requires safeguarding. We adopt the same methods from TGS [52] to obtain the point cloud $\hat{\mathcal{P}}$ in $E_1$. In $E_2$, users can refine this point cloud to select the area they wish to protect. Users can employ text-guided editing techniques like instructP2P [48] or turn to open-source software like MeshLab [5] to further shape and customize the point cloud. This process not only enhances the visual appeal of the point cloud but also embeds a layer of security by aligning it with specific textual instructions or user intents.

## 4.2 Optimizing geometry cloak with view-specific PGD

In this section, by exploiting the vulnerability of TGS [52], we present view-specific PGD to determine geometry cloak $\delta$, which can minimize the distance between reconstructed point clouds in TGS and target pre-defined patterns.

*Geometry Cloak $\delta$.* Geometry cloak $\delta$ is crafted to mislead the TGS into a controlled 3D reconstruction with imperceptible perturbations into single-view images. It minimizes the difference to the target geometry pattern while preserving image fidelity perceptible to the human eye through adversarial training [28].

Specifically, our geometry cloak $\delta$ aims to minimize the Chamfer Distance $\mathcal{L}_{\text{CD}}$ between point cloud $\hat{P}$ from image $\mathcal{I}$ via TGS and target geometry pattern $\mathcal{P}_{\text{tar}}$ in a certain view. The overall objective can be expressed as follows:

$$\delta := \arg \min_{\|\delta\|_\infty \leq \epsilon} \mathcal{L}_{\text{CD}}(\mathcal{E}(\mathcal{I} + \delta), \mathcal{P}_{\text{tar}}), \qquad (3)$$

where $\mathcal{E}$ denotes the network that maps image into point cloud in TGS [52], and $\epsilon$ represents the perturbation budget.

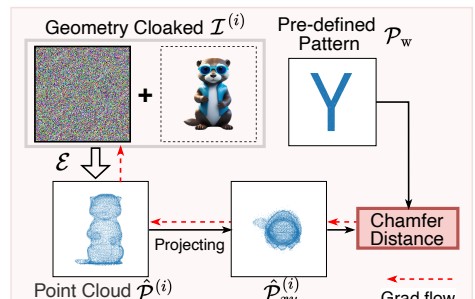

Figure 4: Example of View-specific PGD. We use a 2D point cloud pre-defined pattern $\mathcal{P}_{\text{w}}$ as the target geometry pattern. The watermark is embedded at the viewing direction $\theta = xy$.

*View-specific PGD.* To embed pre-defined patterns in a specific viewing direction, we develop a view-specific PGD. The optimization iteratively adjusts the geometry cloak $\delta$ to make the projected point cloud closer to the pre-defined patterns while keeping the image visually similar to the source. The updates use gradient decent [28] on $\mathcal{L}_{\text{CD}}$ with learning rate $\alpha$, moving the cloaked image towards the target. This iterative process manipulates the TGS's perception while maintaining visual similarity as below:

$$\mathcal{I}^{(i+1)} = \mathcal{I}^{(i)} + \alpha \cdot \text{sgn}(\nabla_{\mathcal{I}} \mathcal{L}_{\text{CD}}(\hat{\mathcal{P}}_\theta^i, \mathcal{P}_{\text{w}})), \qquad (4)$$

where $\hat{\mathcal{P}}_\theta^i = \text{Proj}_\theta(\mathcal{E}(\mathcal{I}^{(i)} + \delta^{(i)}))$, denoting the projected point cloud at viewing direction $\theta$ from cloaked image $(\mathcal{I}^{(i)} + \delta^{(i)})$, and $\mathcal{P}_{\text{w}}$ stands a 2D point cloud watermarks. We encapsulate the geometry cloaking process in Algorithm 1.

---
**Algorithm 1:** Optimizing Geometry Cloak with view-specific PGD
---
**Input:** Input image $\mathcal{I}$, Point cloud encoder $\mathcal{E}$ from TGS, pre-defined pattern $\mathcal{P}_w$
number of steps $N$, step size $\alpha$, perturbation budget $\epsilon$, viewing direction $\theta$
**Output:** Geometry cloaked image $\hat{\mathcal{I}}$
Initialize geometry cloak $\delta \leftarrow 0$, and geometry cloaked image $\hat{\mathcal{I}} \leftarrow \mathcal{I}$
**for** $i \leftarrow 1$ **to** $N$ **do**
$\quad \hat{\mathcal{P}} \leftarrow \mathcal{E}(\hat{\mathcal{I}})$ // Estimate point cloud representations;
$\quad \hat{\mathcal{P}}_\theta \leftarrow \text{Proj}_\theta(\hat{\mathcal{P}})$ // Project the point cloud at view direction $\theta$;
$\quad Loss \leftarrow \mathcal{L}_{\text{CD}}(\hat{\mathcal{P}}_\theta, \mathcal{P}_w)$ // Calculate CD between two 2D point clouds;
$\quad \delta \leftarrow \alpha \cdot \text{sgn}(\nabla_{\hat{\mathcal{I}}} Loss)$ // Update geometry cloak;
$\quad \hat{\mathcal{I}} \leftarrow \hat{\mathcal{I}} + \delta$ // Update cloaked image;
$\quad \hat{\mathcal{I}} \leftarrow \text{clip}(\hat{\mathcal{I}}, \mathcal{I} - \epsilon, \mathcal{I} + \epsilon)$ // Clip cloaked image $\hat{\mathcal{I}}$;
**end**
**Return** $\hat{\mathcal{I}}$
---

### 4.3 Implementation details

Our method uses the PyTorch framework on a single NVIDIA V100 GPU. Our geometry cloak is obtained through optimization (Section 4.2) using projected gradient descent [11]. We adopt a mask version of PGD [28] for calculating geometry cloak, as only the object in the image is used for 3D reconstruction without its background. The input image $\mathcal{I}^{(0)} = \mathcal{I}$ is initialized, and iteratively updated for $N = 100$ steps with a step size of $\alpha = 0.001$. The loss is defined as the Chamfer Distance [1] between the predicted point cloud $\mathcal{P}$ and the target point cloud $\hat{\mathcal{P}}_{tar}$. For pre-defined patterns, we adopt the proposed view-specific PGD as we want to embed the 2D pre-defined patterns into a certain viewing direction $\theta$. For customized patterns, we directly calculate the distance between customized patterns and predicted point clouds, as both are 3D point clouds. The updated image $\mathcal{I}^{(i)}$ is clipped to the valid range $[0, 1]$, and after $N = 100$ iterations, the final image $\hat{\mathcal{I}} = \mathcal{I}^{(N)}$ are obtained as the geometry cloaked output.

## 5 Experiments

### 5.1 Settings

**Dataset.** To evaluate the performance of our method, we conduct experiments on the Google Scanned Objects (GSO) [7] and OmniObject3D (Omni3D) [45] datasets. Both datasets embody a large diversity of view images and provide a rich and varied set of data for assessing the performance of our method. For GSO [7], we select 1 image view for each of the 1030 objects, resulting in 1030 images. For Omni3D [45], we choose 5 image views from each of the 190 classes in Omni3D, resulting in a total of 950 images.

**Baselines.** To investigate the effectiveness of our approach, we evaluate the impact of different perturbations on the reconstructed results by TGS [52]. We experiment with four types of perturbations strategy: (1) **Gauss. noise**, *i.e.*, random Gaussian noise is added to the protected image. (2) **Adv. image**, *i.e.*, adversarial attacks on image feature [9]; (3) **Geometry cloak w/o target**, *i.e.*, adversarial attacks on the point cloud. For Adv. image and geometry cloak w/o target, we directly use the norm value of features [35] as the optimizing loss. (4) **Geometry cloak**. For the geometry cloak, we randomly select letters "A-Z" and numbers "1-9" as the pre-defined pattern.

**Evaluation methodology.** We report the quantitative metric between no perturbation and perturbed reconstructed results. Specifically, for visual quality, we report image similarity metrics: PSNR, SSIM [44], LPIPS [49]. For geometry quality, we report the Chamfer Distance (CD) [1]. We further present the qualitative customized 3D reconstruction by inducing the point from the single-view image into another domain.

Table 1: Quantitative comparison of perturbation strategies We present the outcomes of four distinct perturbation strategies compared to the non-perturbed results. These strategies include Gaussian noise (random Gaussian noise), Adversarial image (perturbing image features), geometry cloak without target, and geometry cloak. We evaluate the image quality metrics (PSNR/SSIM/LPIPS) and geometry quality metric (Chamfer Distance, CD) on the Omni3D [45] and GSO [7] datasets at perturbation budgets of $\epsilon = 2, 4, 8$.

| $\epsilon$ | Perturb. strategy | Omni3D | | | | GSO | | | |
|---|---|---|---|---|---|---|---|---|---|
| | | PSNR $\downarrow$ | SSIM $\downarrow$ | LPIPS $\uparrow$ | CD $\uparrow$ | PSNR $\downarrow$ | SSIM $\downarrow$ | LPIPS $\uparrow$ | CD $\uparrow$ |
| 2 | Gauss. noise | 23.35 | 0.8954 | 0.1141 | 4.913 | 24.24 | 0.9049 | 0.1130 | 5.070 |
| | Adv. image | 19.32 | 0.8424 | 0.1782 | 13.86 | 19.94 | 0.8478 | 0.1821 | 21.89 |
| | Ours w/o target | 11.69 | 0.7513 | 0.2585 | 237.7 | 13.07 | 0.7849 | 0.2499 | 267.5 |
| | Ours | 12.47 | 0.7392 | 0.2669 | 337.0 | 14.06 | 0.7673 | 0.2578 | 352.6 |
| 4 | Gauss. noise | 22.11 | 0.8788 | 0.1313 | 6.424 | 22.66 | 0.8847 | 0.1354 | 7.940 |
| | Adv. image | 19.21 | 0.8417 | 0.1784 | 14.84 | 19.81 | 0.8466 | 0.1827 | 13.60 |
| | Ours w/o target | 11.49 | 0.7513 | 0.2603 | 260.1 | 12.94 | 0.7848 | 0.2504 | 284.8 |
| | Ours | 12.45 | 0.7400 | 0.2665 | 344.9 | 14.04 | 0.7669 | 0.2585 | 356.5 |
| 8 | Gauss. noise | 20.34 | 0.8443 | 0.1651 | 10.20 | 20.68 | 0.8479 | 0.1724 | 13.60 |
| | Adv. image | 19.17 | 0.8412 | 0.1786 | 13.86 | 19.80 | 0.8468 | 0.1824 | 23.58 |
| | Ours w/o target | 11.44 | 0.7515 | 0.2608 | 273.3 | 12.89 | 0.7840 | 0.2508 | 296.3 |
| | Ours | 12.44 | 0.7396 | 0.2664 | 348.4 | 14.02 | 0.7661 | 0.2584 | 358.7 |

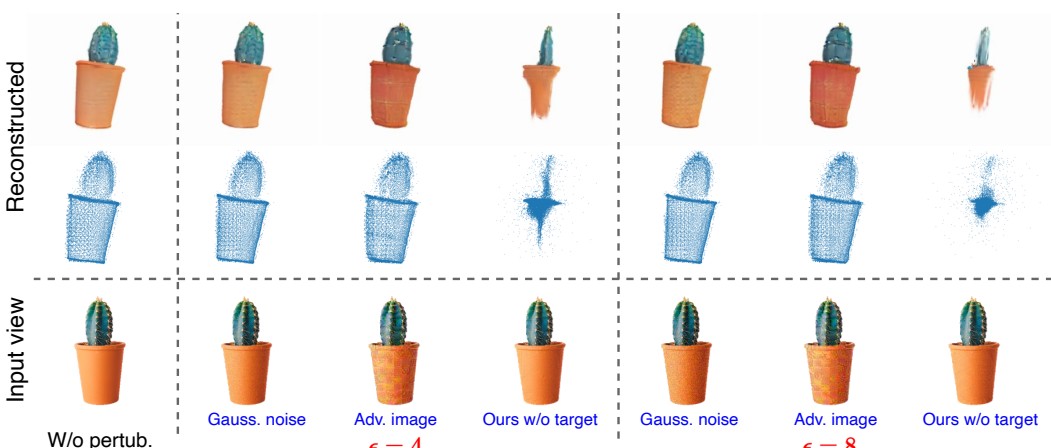

Figure 5: Qualitative reconstructed results with different perturbing strategies. Compare to Gauss. noise and Adv. image, our method can significantly affect the reconstructed results, indicating the explicit geometry features are perturbation-prone during 3D reconstruction.

## 5.2 Experiential results

To showcase the effectiveness of our geometry cloak, we have conducted experiments using various perturbation strategies to assess their capability in preventing 3D reconstruction. We report the visual (PSNR/SSIM/LPIPS) and geometry (CD distance) differences between 3D models reconstructed from perturbed images and unprotected images.

**Perturbing image/geometry features.** To identify the perturbation-prone components of TGS, we compare the perturbation of implicit image features (Adv. image) and explicit geometry features (Ours w/o target) at different intensities norms [35]. As shown in Table 1, the experimental results show that by attacking the image features, the reconstructed visual and geometry quality is only slightly compromised, with the reconstruction results hardly impacted. However, attacking the point cloud resulted in significant changes between the perturbed and the non-perturbed 3D model. We further provide qualitative results in Fig. 5. The image features exhibit strong robustness to perturbations. Even with obvious adversarial perturbations on the image, it is difficult to affect the

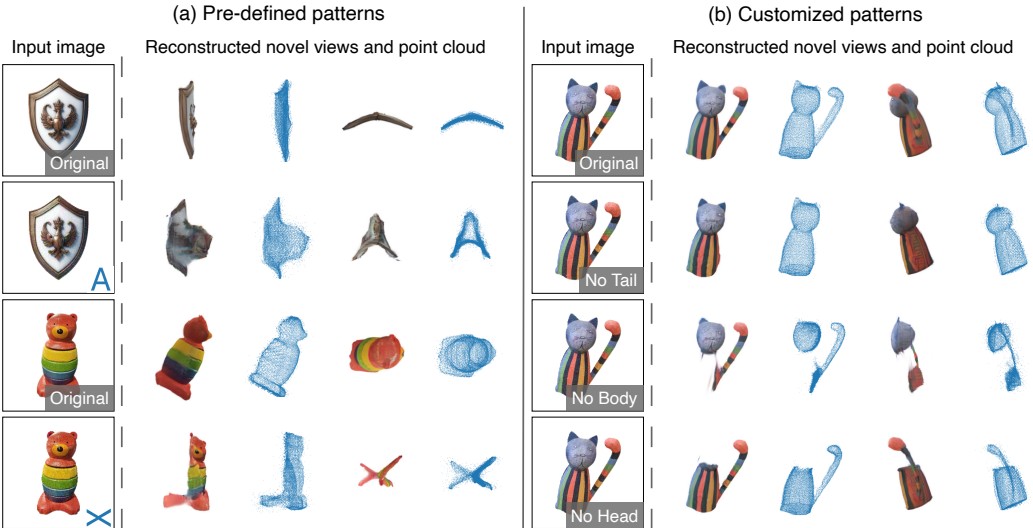

Figure 6: Qualitative results of two different target geometry patterns. (a) Pre-defined patterns: The letters "A" and "X" are used as watermark messages. The embedded watermark can be effectively observed from a certain perspective. (b) Customized patterns: Users can selectively control the parts that need protection, causing the 3D reconstruction of corresponding parts to fail. More qualitative experimental results are provided in the Appendix.

obtained 3D model. On the other hand, attacks on point clouds only require very small, invisible perturbations to change the reconstructed 3D model greatly.

**Pre-defined pattern.** Our method is proposed to perturb explicit geometry and develop a view-specific pre-defined pattern. To evaluate the impacts of our geometry cloak on reconstructing 3D models via TGS [52], we randomly select letters "A-Z" and numbers "1-9" as the pre-defined patterns. As shown in Table 1, we can see that after specifying the target point cloud, the reconstructed 3D model is comprised both in vision and geometry results, making the reconstructed model unusable. We further demonstrate the qualitative results in Fig. 6, showing that the watermark message can be re-emerged in the specific view perspective. This indicates that our method can preserve identifiable embedded information while ensuring the integrity of the 3D model.

**Customized pattern.** We also demonstrate the performance of our method when using customized patterns. In this setting, the users can selectively protect specific parts of the image that need not be reconstructed for customized protection. As shown in Fig. 6, users can choose to remove certain object parts (*e.g.*, tail, body, head). The results indicate that users can effectively influence the reconstruction results with our geometry cloak, further demonstrating that point clouds in TGS are susceptible to perturbation. We provide more results to show that our geometry cloak can be generalized to other GS-based single-view to 3D method [40] in the appendix.

## 6 Conclusion

We present a novel geometry cloaking approach to protect image copyrights from unauthorized 3D reconstruction with Triplane Gaussian Splatting (TGS). By embedding carefully optimized perturbations in the geometry feature space that encodes a customized watermark message, our method forces TGS to fail reconstruction in a distinct way - generating the watermarked pattern. Extensive experiments validate our strategy of focusing perturbations on the geometry components of TGS, which can reliably induce watermarks with invisible perturbations. Our geometry cloaking introduces a novel method for protecting copyrights tailored to the representations of single-view to 3D models.

**Limitations and broader impacts.** The multifaceted nature of copyright protection requires that our method be developed and used responsibly, respecting the delicate balance between innovation and intellectual property rights. Collaboration across technology, legal, and policy sectors is essential to address these complexities.

**Acknowledgement.** This work was done at Renjie's Research Group at the Department of Computer Science of Hong Kong Baptist University. Renjie's Research Group is supported by the National Natural Science Foundation of China under Grant No. 62302415, Guangdong Basic and Applied Basic Research Foundation under Grant No. 2022A1515110692, 2024A1515012822, and the Blue Sky Research Fund of HKBU under Grant No. BSRF/21-22/16.

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

# A   Appendix / supplemental material

## A.1   Additional experimental results.

To better illustrate the effectiveness of our approach (geometry cloak w/o target and geometry cloak), we provide more qualitative experimental results of our geometry cloak.

**Visual results of geometry cloak.**   As shown in Fig. S1 and Fig. S2, geometry cloak w/o target can significantly alter the 3D model generated through TGS [52] without affecting the original visual effect of the image. For the geometry cloak, we embed pre-defined patterns at the top view. As shown in Fig. S3 and Fig. S4, the geometry cloaks are invisible to humans, and the embedded watermarks can be effectively observed from rendered views and reconstructed point clouds.

**Extending to LGM [40].**   As GS-based [18] method requires explicit geometry features to represent 3D scenes, our geometry cloak can also be extended to other GS-based single-view to 3D methods [40]. Fig. S5 and Table S1 provide the qualitative and quantitative results when implementing our method on LGM [40]. Fig. S5 presents the reconstructed views and point cloud via LGM under different perturbating strategies. The reconstructed 3D model is undermined and manipulated via our geometry cloak. In Table S1, we experiment on three default scenes in LGM and report the reconstructed results when applying Gaussian noise and our method to the input views. The results show that our method can effectively disturb the quality of the reconstructed 3D model.

**Combining adv. tri-plane.**   Fig. S6 presents the visual results when combining perturbation on the tri-plane feature and geometry feature. Combining the two does not improve the attack performance, as the tri-plane feature is a robust part of TGS that is difficult to disturb. Future work could focus on studying the components in the 3D reconstruction process that are vulnerable to disturbances.

**Perturbations with smaller/larger budget.**   Fig. S6 provide the visual results when employing a smaller/larger budget . These two figures indicate that our method is insensitive to larger epsilon values, as high-intensity Gaussian noise struggles to disturb the geometric features of the reconstructed results.

**Tendency of performance degradation.**   Fig. S7 illustrates the quality of the reconstructed 3D results under different epsilon intensities. An obvious decrease in reconstruction quality occurs within the 0 to 4 intensity range.

**Convergence status under different budgets.**   Fig. S8 presents the convergence status under different budgets.

**Quality of protected image.**   Fig. S9 shows the quality of the protected image under different perturbation strategies.

**Computational resources.**   Fig. S10 presents the time required to finish protection via our method.

**Viewing direction.**   Fig. S11 illustrates the visual results of embedding a watermark at different angles and observing it from different perspectives.
Table S2 shows the metric results when embedding a watermark at different angles.

**Multi-character as watermarks.**   Fig. S12 presents the results when embedding multi-character as the side view.

**Robustness against image compression.**   Table S2 presents the results when the protected image is modified via common image operations.

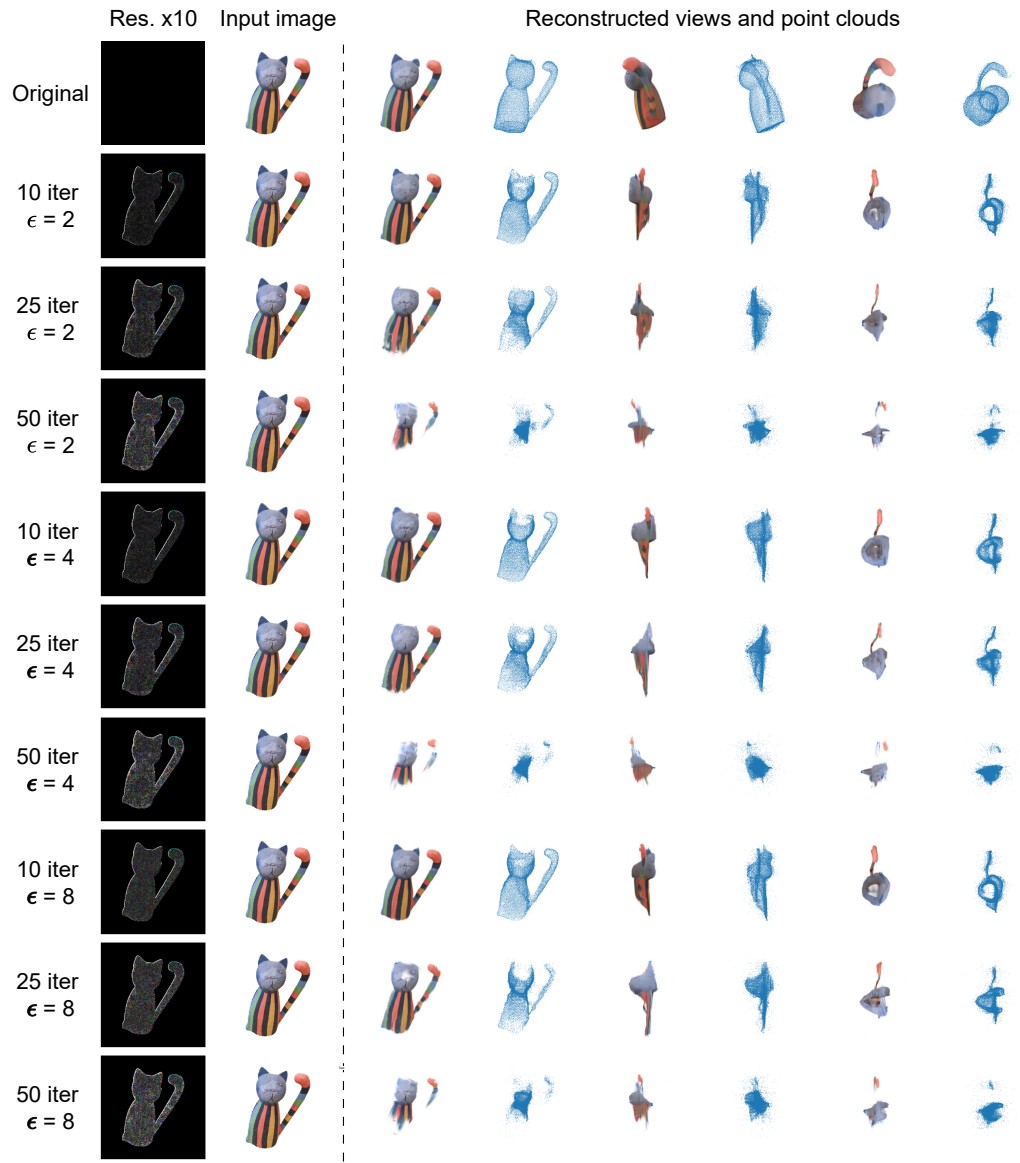

Figure S1: Qualitative results of geometry cloak w/o target. We present the reconstructed views and corresponding point cloud at the front view, side view, and top view.

Table S1: Extending our geometry cloak to LGM [40].

|  | Scene | Reconstructed 3D model | | | Impact on image quality | |
|---|---|---|---|---|---|---|
|  |  | PSNR↓ | LPIPS↑ | CD↑ | $PSNR_{gt}$ ↑ | $LPIPS_{gt}$ ↓ |
| Gau. noise | Anya | 27.97 | 0.0871 | 4.921 | 31.95 | 0.0711 |
|  | Bird | 31.01 | 0.0889 | 3.506 | 29.59 | 0.0907 |
|  | Catstatue | 29.20 | 0.1084 | 1.430 | 29.79 | 0.1042 |
| W/o target | Anya | 14.17 | 0.1676 | 60.52 | 31.84 | 0.0712 |
|  | Bird | 16.02 | 0.1950 | 84.51 | 29.68 | 0.0946 |
|  | Catstatue | 14.65 | 0.2439 | 36.57 | 29.61 | 0.1007 |
| Ours | Anya | 14.04 | 0.2746 | 132.2 | 31.80 | 0.0721 |
|  | Bird | 15.11 | 0.2449 | 185.0 | 29.67 | 0.0971 |
|  | Catstatue | 14.60 | 0.3331 | 125.3 | 29.59 | 0.1039 |

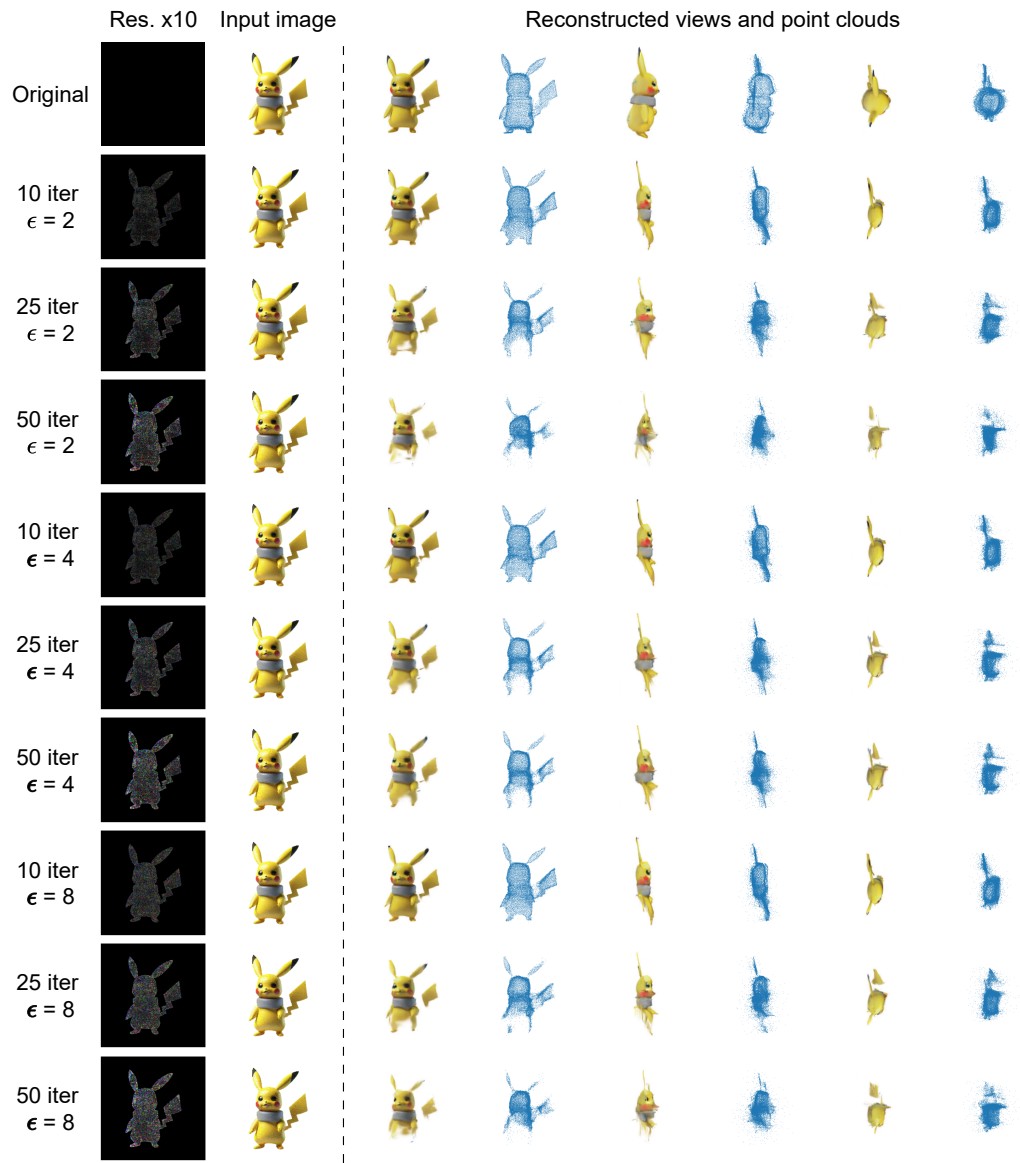

Figure S2: Qualitative results of geometry cloak w/o target. We present the reconstructed views and corresponding point cloud at the front view, side view, and top view.

Table S2: Quantitative results of the 3D reconstructed model derived from a compressed protected image and under various embedding directions.

| | no. comp. | Gaus. noise | | Brightness | | Down-sample | | JGPE | | Embedding direction $\theta$ | | |
|---|---|---|---|---|---|---|---|---|---|---|---|---|
| | | 1.0 | 2.0 | 1.0 | 2.0 | x2 | x4 | 60 | 90 | Front | Side | Top |
| PSNR↓ | 11.05 | 10.97 | 11.21 | 12.92 | 14.72 | 19.12 | 18.73 | 20.13 | 14.71 | 15.4 | 14.37 | 13.02 |
| SSIM↓ | 0.804 | 0.806 | 0.798 | 0.788 | 0.810 | 0.867 | 0.862 | 0.861 | 0.807 | 0.808 | 0.797 | 0.762 |
| LPIPS↑ | 0.194 | 0.194 | 0.197 | 0.186 | 0.162 | 0.113 | 0.122 | 0.111 | 0.158 | 0.170 | 0.172 | 0.213 |
| CD↑ | 155.6 | 118.7 | 93.81 | 9.411 | 25.56 | 41.35 | 10.88 | 14.75 | 42.22 | 138.76 | 150.43 | 193.74 |

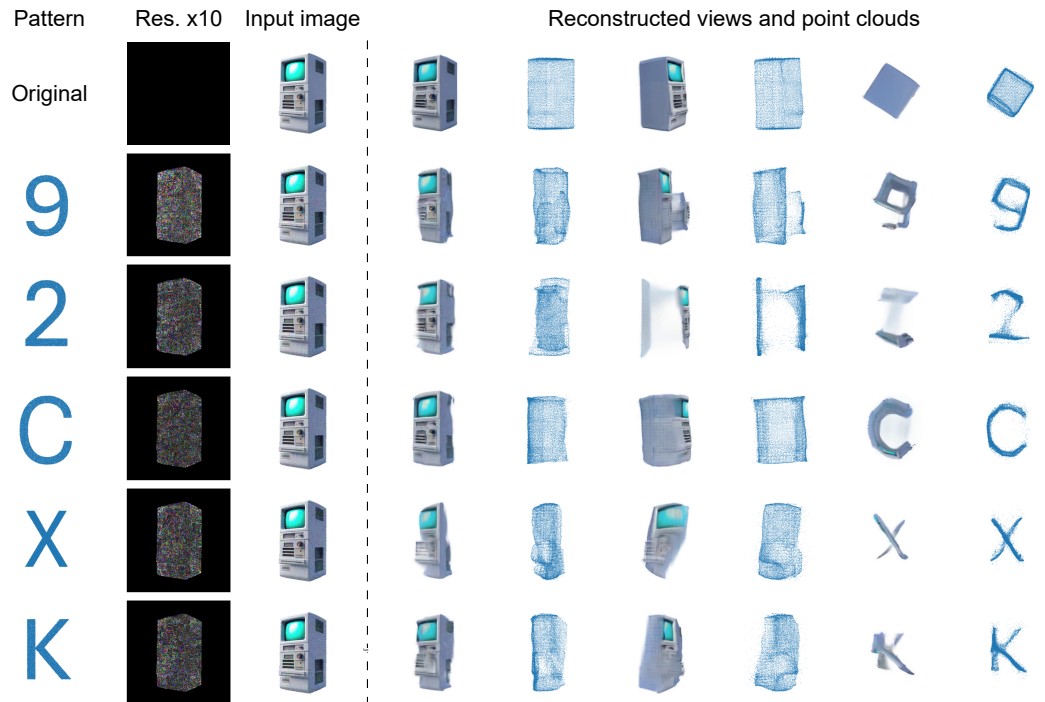

Figure S3: Qualitative results of geometry cloak. We present the reconstructed views and corresponding point cloud at the front view, side view, and top view.

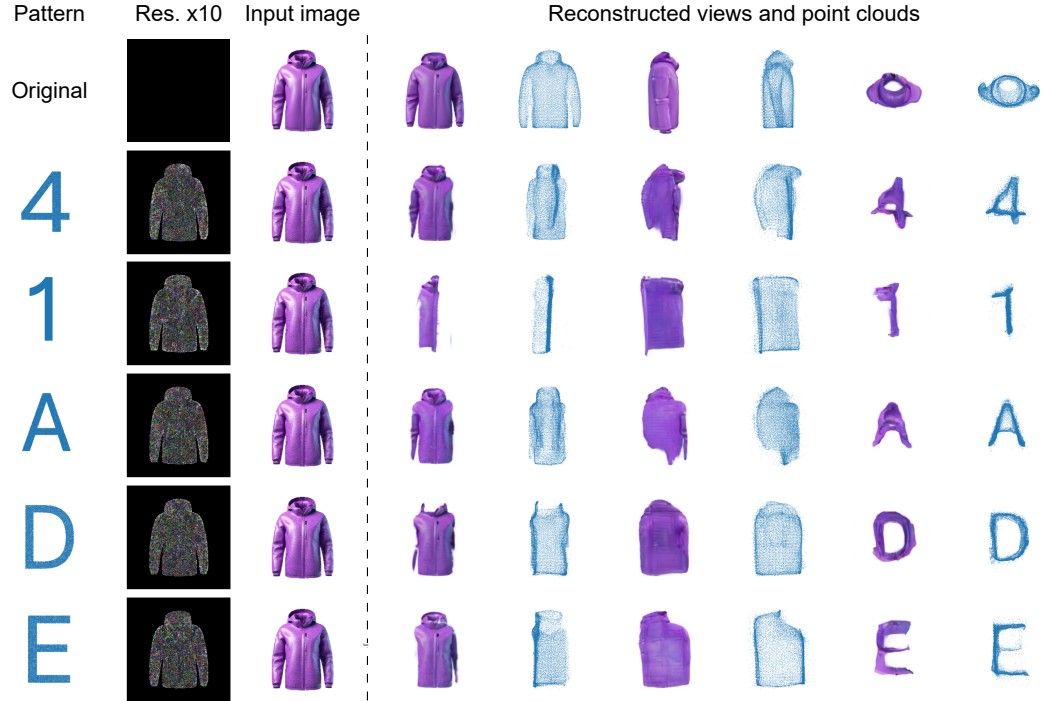

Figure S4: Qualitative results of geometry cloak. We present the reconstructed views and corresponding point cloud at the front view, side view, and top view.

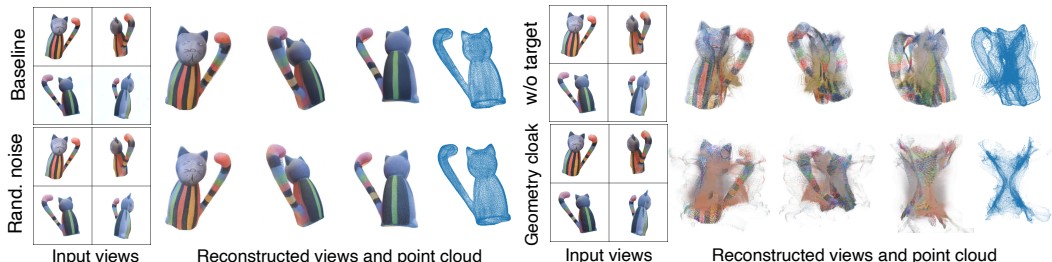

Figure S5: Extending our geometry cloak to LGM [40]. The random noise barely affects the reconstructed results, while our method can effectively manipulate the reconstructed 3D model. Our method can work on LGM as explicit geometry features are vulnerable for the GS-based framework (Please zoom in for the best review).

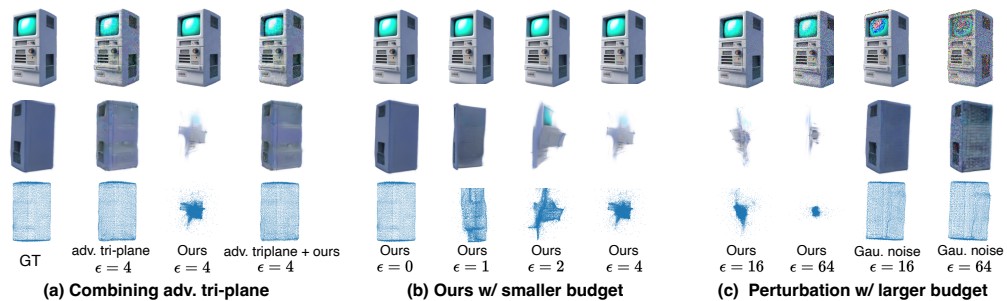

Figure S6: Visual results of different perturbation strategies. **(a)** Different attack strategy combinations. **(b)** The results of our method with smaller $\epsilon$. **(c)** The results of Gaussian noise with higher $\epsilon$ (Please zoom in for best review).

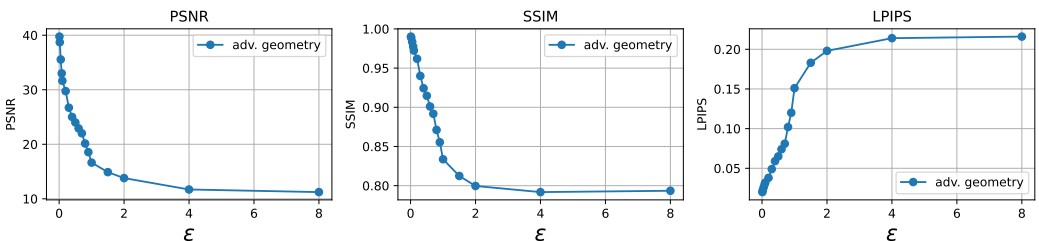

Figure S7: Quality of 3D reconstructed results from protected images under different perturbation budget $\epsilon$.

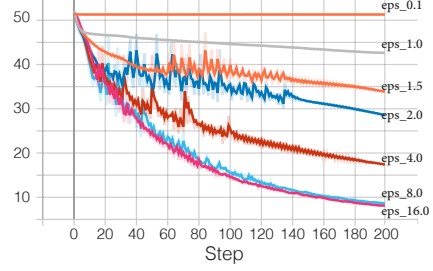

Figure S8: Convergence status under different values of budget $\epsilon$.

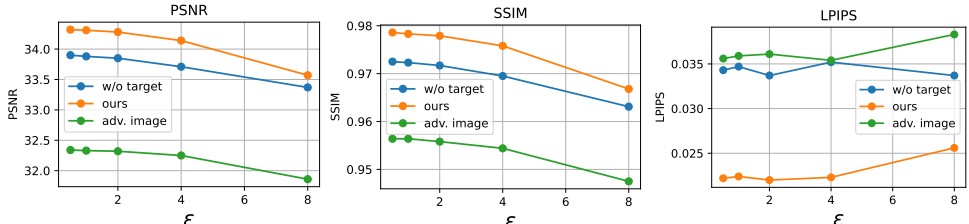

Figure S9: Quality of protected image under different perturbations strategy.

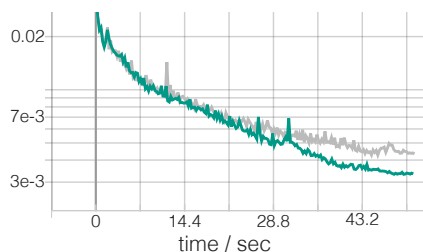

Figure S10: Convergence curve of our method with different $\epsilon$ values (green line $\epsilon = 2$, gray line $\epsilon = 4$). Our method can complete the protection of images within 50 sec.

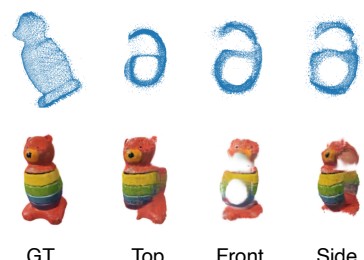

GT  Top  Front  Side

Figure S11: Embedding messages at different view direction $\theta$.

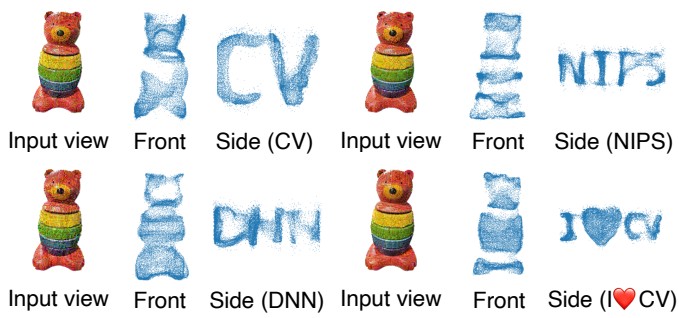

Input view Front Side (CV)  Input view Front Side (NIPS)

Input view Front Side (DNN)  Input view Front Side (I♥CV)

Figure S12: Embedding multi-character as copyrighted messages within side views. Our method still ensures identifiable watermark patterns.

