# OpenReview forum: "Geometry Cloak: Preventing TGS-based 3D Reconstruction from Copyrighted Images"
_NeurIPS.cc/2024/Conference — NeurIPS 2024 poster_

### Official Review · Reviewer_u5Eb · 2024-07-05

**Soundness:** 3
**Presentation:** 2
**Contribution:** 1
**Rating:** 3
**Confidence:** 4

**Summary:**

This paper proposed a new image cloaking approach, which adds adversarial noise on single-view image and makes TGS-based 3D reconstruction fail. This can be served as a watermark for protecting copyright image assets.

**Strengths:**

The topic is popular and needs more investigation by the community. The paper writing is clear.

**Weaknesses:**

1. The main weakness lies in the scope of this work is too limited. The motivation is to protect copyright images from unauthorized 3D reconstruction, however the work only targets at TGS-based reconstruction, which is too limited to have practical effects. I suggest the authors should at least do experiments on other single image 3D reconstruction works, for example LRM[1], Gamba[2] or LGM[3]. This work shows fair results on TGS, but TGS itself is not representative among 3D reconstruction works, and does not yield the best single image 3D reconstruction results.  In practice, we cannot assume unauthorized users will use which image-to-3D model, so it only makes sense when the watermarking / cloaking technique can effectively generalize to shield all those image-to-3D reconstruction algorithms.

2. Some typos.
Line 138, "preventing copyrighted"?


Reference
[1] LRM: Large Reconstruction Model for Single Image to 3D. ICLR 2024
[2] Gamba: Marry Gaussian Splatting with Mamba. https://arxiv.org/abs/2403.18795
[3] LGM: Large Multi-View Gaussian Model for High-Resolution 3D Content Creation. https://arxiv.org/abs/2402.05054

**Questions:**

1. How to specify the observation viewing direction? Is it arbitrarily selected?  How will the specified viewing direction affect reconstruction effects? (e.g. when the observation viewing direction is not good, will the algorithm fail?)

2. The example images shown in the paper use only one character or number for pre-defined patterns. If the watermark information is more complexed and needs multiple characters / numbers, how will the results be like?

3. Regarding weakness, can you discuss how this method can generalize to other single image to 3D reconstruction algorithms?  Is there any more generalized way to do watermarking/cloaking ?

4. In method description, the authors mentioned they use "a mask version of PGD", but details on how to generate the mask is not given in the text.

**Limitations:**

The author has discussed the social impacts. Overall this is a work for copyright protection, which does not have direct negative social impacts.

---

> ### Author Rebuttal · Authors · 2024-08-06
>
> Dear reviewer,
>
> Thank you for your valuable feedback and suggestions.
>
> **Response to W1: Extending to other methods**
>
>
> Our method is designed to utilize the explicit geometry feature in GS-based single views to 3D methods, which is fragile and susceptible to disturbances in the reconstruction process. Thus, our method can work on various GS-based single views to 3D methods as the explicit geometry feature is a necessary element for 3DGS. Before our submission, other GS-based single images to 3D methods were not formally peer-reviewed. While we primarily focus on TGS as it is peer-reviewed during our research, our approach can be extended to other recent methods. To illustrate this, we evaluate our technique on the recently proposed LGM [1], accepted by ECCV 2024. The results are appended in Fig. R1 and Tab. R1 of the rebuttal page. By simply adapting key design elements, our approach shows promising results on LGM as well, underscoring its potential in manipulating other GS-based single-image-to-3D reconstruction methods.
>
>
>
> **Response to W2: Typos**
>
> We appreciate you pointing out these typos. The typos in our paper will be carefully corrected.
>
> **Response to Q1: Observation viewing direction**
>
> 1. In the paper, we present the results from the top view.  In theory, our method can be applied to arbitrarily selected viewing directions. The algorithm will not fail because we are simply mapping the projected point cloud at direction θ, then using this projected point cloud to present the watermark.  We present the results of embedding watermarks at different views in Fig. R5 and Tab. R1 (right). The experimental results indicate that our method can work well when embedding watermarks in different directions.
>
> 2. Our watermarks are embedded at a certain location, which is sufficient for copyright verification of **3D** models. As provided visual results of the watermark from a certain angle from Fig. R5 (Left), viewing the 3D model from other unsuitable perspectives will result in low-quality fragmented information. Besides, to verify copyright, users can verify from any angle, as long as it is identitable and consistent with the pre-embedded watermarks that can be matched.   We will incorporate these results in our next version for better clarification of the property of the geometry cloak.
>
> |                   | Front  | Side   | Top    |
> | ----------------- | ------ | ------ | ------ |
> | PSNR $\downarrow$ | 15.4   | 14.37  | 13.02  |
> | SSIM $\downarrow$ | 0.808  | 0.797  | 0.762  |
> | LPIPS $\uparrow$  | 0.170  | 0.172  | 0.213  |
> | CD $\uparrow$     | 138.76 | 150.43 | 193.74 |
>
>
> **Response to Q2: Multiple characters**
>
> 1. Although the geometry cloak was not specifically designed for multiple characters, our method can still ensure identifiable results when the watermarks are multiple characters. We provide visual results in Fig. R5 (left), which remain promising when there are fewer than four characters.  This suggests that the geometry cloak holds promising potential regarding watermark capacity.
>
>
>
> 2. Besides, compared to previous methods of image cloaking [2,3] against the diffusion model, which can only use artist-style as watermarks, the watermarks embedded via our method are much more identifiable.
>
>
> **Response to Q3: Generalized way to do watermarking/cloaking**
>
> 1. We have discussed how our method can be generalized to other single image to 3D reconstruction algorithms in response to  W1. Preliminary experimental results on LGM [1] are provided in Fig. R1 and Tab. R1, indicating our method is promising in manipulating other GS-based single-image-to-3D reconstruction methods.
>
> 2. To the best of our knowledge, this is the first paper to reveal a vulnerability in the GS-based 3D reconstruction and effectively manipulate the generated results using this vulnerability.   The results generated via GS-based methods can be manipulated through adversarial attacks on geometric features, which is an issue worthy of attention as 3DGS are rather popular today.  More studies should be conducted to protect image/model owners' copyrights and prevent potential malicious attackers.   Our method provides a novel perspective on adversarial perturbations, enabling copyrighted images to prevent misuse by GS-based methods. which could potentially encourage related research in enhancing the robustness of 3D reconstruction and addressing AI privacy/safety issues.
>
>
> **Response to Q4: Details of obtaining masks**
>
> We used SAM to obtain a segmentation of the images to ensure that the perturbations were only added to the objects. We will provide a detailed generation process in our next version of the paper, and experimental codes of geometry cloak will be released to ensure reproducibility.
>
> We look forward to discussing with you in the upcoming discussion phase to clarify things we may neglect.
>
>
> [1] LGM: Large Multi-View Gaussian Model for High-Resolution 3D Content Creation. In ECCV 2024.
>
> [2] Adversarial Example Does Good: Preventing Painting Imitation from Diffusion Models via Adversarial Examples. In ICML 2024.
>
> [3] Glaze: Protecting Artists from Style Mimicry by Text-to-Image Models. In USENIX 2023.

---

> ### Comment · Reviewer_u5Eb · 2024-08-11
>
> Thanks for your rebuttal, which clarifies my questions regarding viewing direction and multiple characters watermarking pattern. However, I still have significant concerns regarding the generalization capabilities of the current version of the work.
>
>
> The authors suggest that their method can generalize to LGM by `simply adapting key elements`. However, the rebuttal materials do not provide sufficient details about the specific algorithmic adjustments required for LGM. LGM employs a fundamentally different architecture to achieve single image to 3D generation. Specifically, it first utilizes off-the-shelf models like ImageDream or MVDream to synthesize multi-view images, followed by an asymmetric U-Net architecture with cross-view self-attentions to construct Gaussian features. Given these foundational differences, it is unclear how the authors' approach could be seamlessly adapted to LGM[1]. For instance, the method described in the paper relies on a point cloud encoder from TGS and uses the Chamfer distance between two-point clouds as a loss to craft adversarial perturbations (c.f. Algorithm 1 and Figure 2). These critical components do not translate directly to LGM[1], as it does not incorporate point cloud encoders.  If authors wants to expand their method to other single image to 3D models, they would have to fundamentally change their motivation statement and algorithm description, which will make a huge modification to their current version of paper.
>
>
> In the rebuttal, the authors mention that their method utilizes `geometry features in GS-based single views to 3D methods`, which I have reservations about this. Their initial claim is that the method is specifically tailored for Triplane Gaussian Splatting.  It is important to note that different approaches to single-image-to-3D conversion employ significantly diverse network architectures (e.g., LRM[2] uses transformers, Gamba[3] uses Mamba), as a result, their `geometry feature` space are highly distinct. Ensuring image copyright protection while considering generalization across these varied models is indeed challenging, and it may not be accurate to make such a broad generalization claim for their methodology design.
>
> TGS[4] is a recently accepted paper in this field, but it is not a representative technique, and its reconstruction results are not among the best. The likelihood of it being widely used in real-world applications remains highly questionable. The key point here is that applying adversarial attacks to **a specific model** is too limited to be considered a broadly useful copyright protection method. Similarly, designing adversarial attacks tailored to a specific method may be too narrow in scope for a NeurIPS paper.
>
> ---
> **Reference**
> [1] LGM: Large Multi-View Gaussian Model for High-Resolution 3D Content Creation. ECCV 2024.
>
> [2] LRM: Large Reconstruction Model for Single Image to 3D. ICLR 2024
>
> [3] Gamba: Marry Gaussian Splatting with Mamba for Single-View 3D Reconstruction. arxiv 2024
>
> [4] Triplane Meets Gaussian Splatting: Fast and Generalizable Single-View 3D Reconstruction with Transformers. CVPR 2024

---

> ### Author Response · Authors · 2024-08-11
> **Thanks for your feedback 1**
>
> We would like to clarify the misunderstanding of the settings of LGM and point cloud encoder.
> LGM[1] and Gamba[3] do have an encoder to get point cloud.
>
> ### **Settings of LGM [1]**
> For LGM, it takes 4 input views obtained from ImageDream/MVDream. Therefore, to verify the effectiveness of our method in undermining the generated 3D results, we directly optimize adversarial perturbations on these four input images.  LGM does have a **point cloud encoder** part to get the point cloud, as 3DGS needs a point cloud to represent the 3D scene. Specifically, please refer to Line 109 of LGM/core/model.py: https://github.com/3DTopia/LGM/blob/main/core/models.py
> ```
> 109        pos = self.pos_act(x[..., 0:3]) # [B, N, 3]
> 110        opacity = self.opacity_act(x[..., 3:4])
> 111        scale = self.scale_act(x[..., 4:7])
> 112        rotation = self.rot_act(x[..., 7:11])
> 113        rgbs = self.rgb_act(x[..., 11:])
> 114        gaussians = torch.cat([pos, opacity, scale, rotation, rgbs], dim=-1) # [B, N, 14]
> ```
> In our rebuttal settings, we directly target and perturbed the pos property of Gaussians,
> Line 109 pos = self.pos_act(x[..., 0:3]) # [B, N, 3],
> which is the center of 3DGS. (i.e., point cloud).
>
>
> ### **Adapting to LRM [2] and Gamba [3]**
>
> For NeRF-based methods like LRM [2], as it does not require explicit geometry features, our method may not be suitable. We didn't experiment with Gamba as it has not been formally peer-reviewed.  However, it also has a point cloud encoder to estimate the point cloud (Line 130, position).
>
> https://github.com/kyegomez/Gamba/blob/main/gamba_torch/main.py
> ```
> 130     # Position, opacity, color
> 131     position = self.position_layer(features)
> 132     opacity = self.opacity_layer(features)
> 133     color = self.color_layer(features)
> ```
> Gamba is still required to estimate the point cloud.
>
> Thus, we do not need to change our motivation statement and algorithm description. The GS-based method must have a point cloud encoder to represent the 3D scene.
>
>
> [1] LGM: Large Multi-View Gaussian Model for High-Resolution 3D Content Creation. ECCV 2024.
>
> [2] LRM: Large Reconstruction Model for Single Image to 3D. ICLR 2024
>
> [3] Gamba: Marry Gaussian Splatting with Mamba for Single-View 3D Reconstruction. arxiv 2024

---

> ### Author Response · Authors · 2024-08-11
> **Thanks for your feedback 2**
>
> Dear reviewer,
>
> Thanks for your valuable feedback.
>
> Our experimental findings indicate that explicit geometry features can be effectively utilized to protect ownership in GS-based tasks, despite TGS being a recent development. Due to the explicit feature of GS, GS-based image-to-3D methods inherently require similar explicit geometry features. This insight suggests that our approach can be extended to these methods, potentially enhancing security and ownership protection in single-image-to-3D applications and inspiring future research in this area.
>
> We strongly agree with you on the concerns of the generalization ability. We will incorporate these additional experiments about our effectiveness on other GS-based approaches in the final version based on your valuable suggestions. We look forward to addressing your further concerns during this discussion.

---

### Official Review · Reviewer_rPf4 · 2024-07-08

**Soundness:** 3
**Presentation:** 4
**Contribution:** 4
**Rating:** 7
**Confidence:** 4

**Summary:**

This paper proposes a novel method to protect copyrighted images from unauthorized 3D reconstruction using Triplane Gaussian Splatting (TGS). This topic is very interesting and highly valuable for preventing the misuse of copyrighted images. Their proposed method achieves protection by incorporating invisible geometry perturbations into the image, which is easy to deploy with less cost. The extensive experimental results have demonstrated the effectiveness of their approach in preventing unauthorized 3D reconstructions while embedding a verifiable pattern. This paper can provide a contribution to addressing the growing issue of image abuse and raise the community's attention to this issue.

**Strengths:**

1. The motivation of this paper is well-stated and significant, and the paper is well-written. The idea of using geometry cloaks to protect images from unauthorized 3D reconstruction is novel. This paper is an attempt to address the issue of image abuse in 3D reconstruction. Such an issue has not received sufficient attention, but it is very important, especially when 3D reconstruction technologies are becoming more accessible. This paper can also raise the community's awareness of this issue.

2. Besides significantly distorting the 3D reconstruction from the protected images, the idea of embedding identifiable patterns into output renderings is interesting. This allows the image owners to determine whether the generated 3D models have used their copyrighted images. This traceability property can enhance the practicality of the method for digital copyright protection.

3. The paper provides extensive experimental results, validating the effectiveness of the proposed approach across different datasets and perturbation strategies. This thorough evaluation helps in building confidence in the robustness and reliability of the proposed method. The approach is scalable and can be applied to a wide range of images without requiring significant computational resources. This enables artists and content creators to safeguard from being misused illegally.

**Weaknesses:**

1. The paper lacks quantitative metrics comparing the similarity between protected and unprotected images, such as PSNR, which would provide a more comprehensive evaluation of the method's impact on image quality.

2. The importance of the view-specific PGD angle has not been thoroughly explored. It would be beneficial to investigate whether the method is effective from different angles, not just the top view. Besides, the effectiveness of combined attacks on point cloud and triplane latent features should be explored to determine if they provide better protection.

3. Minor Issues:
    a). Add a period at the end of "Quantitative comparison of perturbation strategies" in the Table 1 caption.
    b). Replace "no perturbation" with "not perturbed" (L255 P7).
    c). Correct the citation format "et al" to "et al." (L95 P3).

**Questions:**

Please see the weakness.

**Limitations:**

The authors have adequately addressed the limitations

---

> ### Author Rebuttal · Authors · 2024-08-06
>
> Dear reviewer,
>
> Thank you for your recognize and valuable suggestions.
>
> **Response to W1: Impact on image quality**
>
> Our geometry cloak is designed to be invisible so legitimate users can have visually consistent results with the original image quality.  All perturbations are controlled within a certain budget $\epsilon$. Hence, we neglect to provide the impact on image quality.   For a more comprehensive evaluation of the method's impact on image quality, we provide quantitative metrics of the similarity between protected and unprotected images as below:
>
> |                 | $\epsilon$ | PSNR$\uparrow$  | SSIM$\uparrow$   | LPIPS$\downarrow$   |
> |-----------------|-----|-------|--------|---------|
> |Ours                 | 2   | 34.28 | 0.9779 | 0.0220  |
> |             | 4   | 34.14 | 0.9758 | 0.0223  |
> |                 | 8   | 33.57 | 0.9668 | 0.0256  |
> |Ours w/o target                 | 2   | 33.85 | 0.9717 | 0.0337  |
> |  | 4   | 33.71 | 0.9695 | 0.0352  |
> |                 | 8   | 33.37 | 0.9631 | 0.0337  |
> | Adv. image                | 2   | 32.32 | 0.9558 | 0.0361  |
> |     | 4   | 32.25 | 0.9544 | 0.0354  |
> |                 | 8   | 31.86 | 0.9475 | 0.0383  |
>
> Compared to previous methods of adversarial attacks on image features, our proposed geometry cloak ensures higher invisibility while effectively disturbing the reconstruction results. Besides TGS, our method also keeps the invisibility for LGM[1], as shown in Tab. R1 of the rebuttal page.
>
>
> **Response to  W2: Viewing direction and Adv. triplane**
>
> (View-specific PGD) We conduct experiments from other perspectives (side/front). The experimental results indicate that the reconstructed results can still be effectively manipulated from these perspectives, demonstrating the effectiveness of our method. We also provide results of embedding multiple letters further to demonstrate the view-specific PGD performance (Fig. R5 and Tab. R1).
>
> | | Front  | Side   | Top    |
> |----------------------------|--------|--------|--------|
> | PSNR $\downarrow$          | 15.4   | 14.37  | 13.02  |
> | SSIM $\downarrow$          | 0.808  | 0.797  | 0.762  |
> | LPIPS $\uparrow$           | 0.170  | 0.172  | 0.213  |
> | CD $\uparrow$              | 138.76 | 150.43 | 193.74 |
>
> (Effectiveness of  point cloud  + triplane latent features )  Following your suggestions, we conduct experiments by combining attacking point cloud and triplane latent features. The results in Fig. R2 (a) indicate that attacking only the tri-plane will affect the visual quality of the reconstruction. There is no significant change after combining attacks on the point cloud. Future work could focus on studying the components in the 3D reconstruction process that are vulnerable to disturbances.
>
>
> **Response to W3: Minor Issues**
>
> We appreciate you pointing out these typos. These typos in our paper will be carefully corrected in the next version of our paper.
>
>
> [1]  LGM: Large Multi-View Gaussian Model for High-Resolution 3D Content Creation. in ECCV 2024.

---

> > ### Comment · Reviewer_rPf4 · 2024-08-10
> >
> > Thank you for your response. All of my concerns are addressed. The experiments of adapting to other methods should be incorporated into the main paper. The figures in the rebuttal for muli-char embedding are impressive. Thus, I will keep my original rating.

---

> > > ### Author Response · Authors · 2024-08-11
> > > **Thanks for your feedback**
> > >
> > > Comment: Dear Reviewer,
> > >
> > > We are very grateful for your recognition. We will incorporate the experimental results into the main paper following your valuable suggestions.
> > >
> > > Best regards,
> > > Authors of #1185

---

### Official Review · Reviewer_inRn · 2024-07-11

**Soundness:** 3
**Presentation:** 3
**Contribution:** 3
**Rating:** 7
**Confidence:** 4

**Summary:**

The paper introduces a novel approach to protect copyrighted images from unauthorized 3D reconstructions using Triplane Gaussian Splatting (TGS). The method involves embedding invisible geometry perturbations, termed "geometry cloaks," into images. These cloaks cause TGS to fail in a specific way, generating a recognizable watermark, thus protecting the original content.

**Strengths:**

1. The concept of protecting images against unauthorized 3D reconstruction using geometric perturbations is novel.
2. This paper is well-detailed and well-written.

**Weaknesses:**

1. The approach is tailored specifically to TGS-based 3D reconstruction. Can the geometry cloak technique be adapted or extended to protect against other 3D reconstruction methods beyond TGS?
2. What are the potential impacts on image quality for legitimate uses when these cloaks are applied?
3. The optimization process for generating geometry cloaks might introduce significant computational overhead, which is not thoroughly discussed in the paper.

**Questions:**

See Weaknesses

**Limitations:**

Mentioned in the manuscript.

---

> ### Author Rebuttal · Authors · 2024-08-06
>
> Dear reviewer,
>
> Thank you for your recognition and valuable suggestions.
>
> **Response to W1: Extending to other methods**
>
> Our method is designed to utilize the explicit geometry feature in GS-based single views to 3D methods, which is fragile and susceptible to disturbances in the reconstruction process. Thus, our method can work on various GS-based single views to 3D methods as the explicit geometry feature is a necessary element for 3DGS. Before our submission, other GS-based single images to 3D methods were not formally peer-reviewed. While we primarily focus on TGS as it is peer-reviewed during our research, our approach can be extended to other recent methods. To illustrate this, we evaluate our technique on the recently proposed LGM [1], accepted by ECCV 2024. The results are appended in Fig. R1 and Tab. R1 of the rebuttal page. By simply adapting key design elements, our approach shows promising results on LGM as well, underscoring its potential in manipulating other GS-based single-image-to-3D reconstruction methods.
>
>
> **Response to W2: Impact on image quality**
>
> Our geometry cloak is designed to be invisible so legitimate users can have visually consistent results with the original image quality.  All perturbations are controlled within a certain budget $\epsilon$. Hence, we neglect to provide the impact on image quality.   For a more comprehensive evaluation of the method's impact on image quality, we provide quantitative metrics of the similarity between protected and unprotected images as below:
>
> |                 | $\epsilon$ | PSNR$\uparrow$  | SSIM$\uparrow$   | LPIPS$\downarrow$   |
> |-----------------|-----|-------|--------|---------|
> | Ours                | 2   | 34.28 | 0.9779 | 0.0220  |
> |            | 4   | 34.14 | 0.9758 | 0.0223  |
> |                 | 8   | 33.57 | 0.9668 | 0.0256  |
> |Ours w/o target                | 2   | 33.85 | 0.9717 | 0.0337  |
> |  | 4   | 33.71 | 0.9695 | 0.0352  |
> |                 | 8   | 33.37 | 0.9631 | 0.0337  |
> | Adv. image               | 2   | 32.32 | 0.9558 | 0.0361  |
> |     | 4   | 32.25 | 0.9544 | 0.0354  |
> |                 | 8   | 31.86 | 0.9475 | 0.0383  |
>
> Compared to previous methods of adversarial attacks on image features, our proposed geometry cloak ensures higher invisibility while effectively disturbing the reconstruction results. Besides TGS,  our method yields consistent invisible results in LGM[1], demonstrating the generality and concealment of our method (as shown in Tab. R1 of the rebuttal page).
>
>
> **Response to W3: Computational resources**
>
> Our method only optimizes the invisible cloak, which does not require a lot of computational resources (~8GB GPU memory). We provide the convergence curve of the loss during the optimization process on the rebuttal page.  In less than 50 secs, with a single V100 GPU, we can achieve protection for each image. The computational resources will be incorporated in the next version of our paper.
>
>
> [1] LGM: Large Multi-View Gaussian Model for High-Resolution 3D Content Creation. In ECCV 2024.

---

> > ### Comment · Reviewer_inRn · 2024-08-10
> >
> > Thank you for the thorough response. Your detailed explanations have resolved most of my concerns. Given the method's demonstrated extensibility and invisibility, I will be increasing my score.
> >
> > One additional suggestion: it may be beneficial to include a pixel-wise difference map between protected and unprotected images.

---

> > > ### Author Response · Authors · 2024-08-10
> > > **Thanks for your feedback**
> > >
> > > Dear Reviewer,
> > >
> > > We are very grateful for your recognition. We will provide pixel-wise difference maps for protected and unprotected images in the final version based on your valuable suggestions.
> > >
> > > Best regards,
> > >
> > > Authors of #1185

---

### Official Review · Reviewer_ykxU · 2024-07-11

**Soundness:** 4
**Presentation:** 4
**Contribution:** 4
**Rating:** 8
**Confidence:** 5

**Summary:**

The paper introduces a novel image protection approach called "Geometry Cloak" to prevent unauthorized 3D model generation from copyrighted images using single-view 3D reconstruction methods like Triplane Gaussian Splatting (TGS). The Geometry Cloak embeds invisible geometry perturbations into images, which are revealed as a customized message when TGS attempts 3D reconstructions, thus acting as a watermark for copyright assertion.

**Strengths:**

1. This paper raises a novel question, namely how to protect the copyright in the process of image to 3D, which is very meaningful.
2. The presentation of this paper is very clear and easy to understand.
3. A view-specific PGD strategy is proposed to optimize geometry cloak, which is simple but effective.
4. The authors conduct experiments on two 3D datasets and various types of patterns, and verify the effectiveness of the experimental results via sufficient ablation experiments and visualization results.

**Weaknesses:**

1. Can the proposed method be extended to other image to 3D models, such as LRM [1], LGM [2]? The author could introduce the advantages of using TGS instead of other image to 3D models in terms of generation speed and quality, so that the readers can better understand its task scenario.

[1] Lrm: Large reconstruction model for single image to 3d. In ICLR 2024.

[2] Large multi-view gaussian model. In ECCV 2024.

2. The robustness of the proposed method should be verified. For example, when Gaussian noise and JPEG compression are added to the protected image, can it still resist illegal theft?

**Questions:**

Please refer to the weakness.

**Limitations:**

The authors have clearly presented their limitations.

---

> ### Author Rebuttal · Authors · 2024-08-06
>
> Dear reviewer,
>
> Thank you for your recognition and valuable suggestions.
>
> **Response to W1: Extending to other methods**
>
> Our method is designed to utilize the explicit geometry feature in GS-based single views to 3D methods, which is fragile and susceptible to disturbances in the reconstruction process. Thus, our method can work on various GS-based single views to 3D methods as the explicit geometry feature is a necessary element for 3DGS. Before our submission, other GS-based single images to 3D methods were not formally peer-reviewed. While we primarily focus on TGS as it is peer-reviewed during our research, our approach can be extended to other recent methods. To illustrate this, we evaluate our technique on the recently proposed LGM [1], accepted by ECCV 2024. The results are appended in Fig. R1 and Tab. R1 of the rebuttal page. By simply adapting key design elements, our approach shows promising results on LGM as well, underscoring its potential in manipulating other GS-based single-image-to-3D reconstruction methods. For NeRF-based methods like LRM [2],  our work could inspire more research to study the vulnerability layers in these frameworks.
>
>
> TGS combines explicit geometry features and implicit tri-plane representations to achieve accurate and detailed reconstructions, representing a cutting-edge approach to 3D object reconstruction from **single-view images**. Before our submission, LGM had not been officially peer-reviewed. Thus, we choose TGS as our experiment subject to verify the effectiveness of the geometry cloak.  However, this does not affect the generality of our approach in GS-based methods. As discussed in the paper, GS-based methods require explicit geometric features to represent 3D models. The process of obtaining this geometric feature is easily manipulable with proper adversarial perturbations. We will extend the experimental results in LGM to our paper to further clarify the effectiveness of the geometry cloak.
>
>
>
>  **Response to W2: Robustness against image compression**
>
> To verify the robustness of our method, we experiment with some common image operations, including Gaussian noise, JPEG, etc. Under mild operations, the geometry cloak cannot be removed, and the geometric features of the generated 3D result are still disturbed. With higher operations, the geometry cloak can be affected. However, the quality of the view image has been severely compromised, leading to poor visual effects in the reconstructed results, making these protected images unusable.
>
> |  | no. Comp. | Noise |  |  | JPEG |  |
> |:---:|:---:|:---:|:---:|:---:|:---:|:---:|
> |  | - | 5.0 | 10.0 |  | 60 |  90 |
> | PSNR $\downarrow$ | 11.05 | 10.97 | 11.21 |  | 20.13 | 14.71 |
> | SSIM $\downarrow$ | 0.804 | 0.806 | 0.798 |  | 0.861 | 0.807 |
> | LPIPS $\uparrow$ | 0.194 | 0.194 | 0.197 |  | 0.111 | 0.158 |
> | CD $\uparrow$ | 155.6 | 118.7 | 93.81 |  | 14.75 | 42.22 |
>
> We present more results about robustness against common image operations in Tab. R1 of the rebuttal page.
>
>
> [1] LGM: Large Multi-View Gaussian Model for High-Resolution 3D Content Creation. In ECCV 2024.
>
> [2] LRM: Large reconstruction model for single image to 3d. In ICLR 2024.

---

> > ### Comment · Reviewer_ykxU · 2024-08-10
> >
> > Thank you for your detailed response. My concerns are well-addressed. Specifically, this method exhibits notable effectiveness across various GS-based single-image-to-3D approaches, proving its broader applicability. Besides, from experimental results in the rebuttal, this paper also demonstrates that some complex patterns can also be efficiently generated for copyright protection. Considering its rebuttal and the two additional merits, I will increase my rating to 8.

---

> > > ### Author Response · Authors · 2024-08-11
> > > **Thanks for your feedback**
> > >
> > > Dear Reviewer,
> > >
> > > We are very grateful for your recognition. We will integrate these results into the next version of our paper based on your valuable feedback.
> > >
> > > Best regards,
> > > Authors of #1185

---

### Official Review · Reviewer_kd52 · 2024-07-11

**Soundness:** 3
**Presentation:** 3
**Contribution:** 2
**Rating:** 6
**Confidence:** 4

**Summary:**

This paper presents a method for copyrights in 3D reconstruction, specifically targeting novel-view synthesis, rather than traditional 2D images. Recently, advancements in 3D reconstruction have been driven by neural radiance fields (NeRFs) and 3D Gaussian splatting (3D GS), both of which maintain 3D consistency. These methods primarily focus on learning RGB values from multiple posed images. Notably, recent developments have shown that single-view 3D reconstruction can be achieved with the support of pre-trained diffusion-based generative models.

Unlike 2D images, privacy preservation in 3D reconstruction has received less attention due to its recent development. This paper introduces a geometry cloak that perturbs 3D point cloud representations instead of 2D images. This perturbation prevents 3D reconstruction from 2D posed images without degrading the visual quality of the 2D images.

The technique is particularly supportive of 3D GS, enabling real-time novel-view synthesis, a capability not supported by existing NeRFs. To the best of our knowledge, this paper is the first to propose an adversarial attack to preserve copyright in 3D GS from a 3D reconstruction perspective.

**Strengths:**

1. This paper addresses contemporary issues regarding copyright protection in 3D reconstruction. Utilizing the framework of Tri-Plane Gaussian Splatting (TGS), which encodes posed images into tri-plane representations and 3D point clouds before generating 3D Gaussian splatting, the paper proposes a novel perturbation method. This method degrades the quality of 3D geometry without affecting the visual quality of 2D images.

2. This demonstrates that the proposed perturbation method significantly impacts the performance of 3D reconstruction, regardless of the degree of perturbation, unlike simple noise injections. It also shows that this method is robust and not sensitive to hyper-parameter variations.

**Weaknesses:**

1. This paper heavily relies on the prior work of Tri-Plane Gaussian Splatting (TGS), which explicitly represents 3D point clouds to enhance geometric properties and employs tri-plane representation to encode 3D Gaussians. Without leveraging TGS, this study would not effectively address copyright issues in 3D reconstruction. This dependency indicates that the proposed approach is not applicable to a wide range of novel-view synthesis techniques, highlighting a limitation in its generalizability.

2.  While this paper demonstrates that the proposed method is insensitive to the degree of perturbation, it does not adequately explain this phenomenon in the context of privacy preservation. In the past, understanding the tendency of performance degradation is crucial and be well-explained through probabilistic analysis for the extent of perturbation. However, the experimental results presented in this paper do not support this concept.

**Questions:**

1. Could you show the slope of performance deviation when $\epsilon$ is less than 2? When $\epsilon = \{0, 1\}$, does the proposed method also exhibit a negative slope of performance in terms of PSNR, SSIM, and LPIPS?

2. Could you show the performance degradation of random noise when $\epsilon$ increases beyond 8? It should demonstrate that the proposed approach is more beneficial under strong perturbation. While $\epsilon=8$ indicates color disturbance, the context and shape in the 2D images do not change.

3. Could you present the ablation study on the influences of PGD?

**Limitations:**

Please refer to weakness.

---

> ### Author Rebuttal · Authors · 2024-08-06
>
> Dear reviewer,
>
> We express our gratitude for your recognition and valuable suggestions.
>
>
> **Response to W1: Extending to other methods**
>
> Our method is designed to utilize the explicit geometry feature in GS-based single views to 3D methods, which is fragile and susceptible to disturbances in the reconstruction process. Thus, our method can work on various GS-based single views to 3D methods as the explicit geometry feature is a necessary element for 3DGS. Before our submission, other GS-based single images to 3D methods were not formally peer-reviewed.  While we primarily focus on TGS as it is peer-reviewed during our research, our approach can be extended to other recent methods. To illustrate this, we evaluate our technique on the recently proposed LGM [1], accepted by ECCV 2024.  The results are appended in Fig. R1 and Tab. R1 of the rebuttal page. By simply adapting key design elements, our approach shows promising results on LGM as well, underscoring its potential in manipulating other GS-based single-image-to-3D reconstruction methods.
>
>
>
>
>
> **Response to W2:  Tendency of performance degradation**
>
> We are very grateful for you pointing out this phenomenon.
>
> Tab.1 in the main paper aims to demonstrate that even with larger budgets,  other perturbation methods (Gaussian noise and adv. image features)  still do not effectively undermine the reconstruction results.  To understand the tendency of perturbation, we provide more experimental results under a wider range of budget $\epsilon$.
>
> | $\epsilon$ | PSNR$\downarrow$  | SSIM$\downarrow$   | LPIPS$\uparrow$ |
> | ---------- | ----- | ------ | ----- |
> | 0.5        | 24.00 | 0.9147 | 0.065 |
> | 0.8        | 20.14 | 0.8711 | 0.102 |
> | 1.0        | 16.64 | 0.8337 | 0.151 |
> | 1.5        | 14.89 | 0.8124 | 0.183 |
> | 2.0        | 13.80 | 0.7996 | 0.198 |
> | 4.0        | 11.71 | 0.7918 | 0.214 |
> | 8.0        | 11.23 | 0.7935 | 0.216 |
> | 16.0       | 11.20 | 0.7914 | 0.218 |
>
> Besides this, we provide visual results of perturbed results in Fig. R2 (b-c) and curves for different  $\epsilon$  in Fig. R3 of the rebuttal page.
>
> The experimental results reveal a sensitivity to smaller budget $\epsilon$ (<2) values, while larger budget $\epsilon$ (>2) values demonstrate insensitivity due to the already severe disruptive effects.  We recognize and thank your suggestions that understanding the tendency of performance degradation is crucial; these results will be extended to the next version of our paper.
>
>
> **Response to  Q1:  Slope of performance deviation**
>
> We provide the slope of performance deviation in Fig. R2 of the rebuttal page, which also exhibits a negative performance slope.
>
>
> **Response to  Q2: Results under stronger perturbation**
>
> We provide visual results when a random noise ($\epsilon$ =16, 32) is applied in Fig R2 (c)  of the rebuttal page.  Even with this larger budget $\epsilon$, The geometry pattern of the reconstructed 3D model has no obvious change.  This indicates that the process of obtaining geometry features through TGS is robust to noise. We will incorporate these results into our paper.
>
> Our method shows unobvious changes in disturbance results after $\epsilon$ > 4 (as discussed in W1). We present the performance slope in Fig. R3  of the rebuttal page.
>
>
> **Response to Q3: Ablation study of PGD**
>
> We demonstrate the convergence of loss under different values of epsilon in PGD (Fig. R3). We also provide the results of PGD attacks from different perspectives (Fig. R5 and Tab. R1) and multiple characters as watermarks (Fig. R5) on the rebuttal page. We look forward to discussing with you in the upcoming discussion phase to clarify things we may neglect.
>
> [1] LGM: Large Multi-View Gaussian Model for High-Resolution 3D Content Creation. In ECCV 2024.

---

> ### Comment · Reviewer_kd52 · 2024-08-11
> **Response to the author's rebuttal**
>
> I am satisfied with the authors' response and appreciate their effort to address my concerns regarding the effectiveness of noise perturbation depending on the privacy budget.
>
> Additionally, It is impressive that performance decreases as perturbation increases.
> While the authors notice that noise to the Tri-Plane do not produce the expected results, the reason for this phenomena remains unexplained. Although they have indicated this as future works, understanding the phenomena seems crucial in 3D reconstruction since learned Tri-Plane appears to also contain geometry information. Given these issue, I maintain my original score.

---

> > ### Author Response · Authors · 2024-08-12
> > **Thanks for your feedback**
> >
> > Dear reviewer kd52,
> >
> > We are very grateful for your recognition. We will integrate the results of the privacy budget into the final version of our paper based on your valuable feedback.
> >
> > 3DGS-based single-view to 3D methods are novel mechanisms recently proposed. The properties of such mechanisms are still under exploration.  One possible explanation for this phenomenon could be that the feature of Tri-Plane is high-dimensional and implicit, while the geometric features (point cloud) are lower-dimensional and explicit. Explicit point clouds directly represent the attributes of the Gaussian (position of Gaussian). This discrepancy may make it easier to obtain appropriate perturbations when dealing with point clouds. We appreciate your insights, and we will incorporate all your valuable suggestions into our final paper.
> >
> > Best regards,
> >
> > Authors of #1185

---

### Author Rebuttal · Authors · 2024-08-06

Dear reviewers,

We would like to thank all the reviewers for their time and for writing thoughtful reviews of our work.

In this work, we introduce the geometry cloak, which can effectively manipulate the process of 3DGS-based single-image to 3D methods by adding invisible perturbation. We reveal that the explicit geometric features are vulnerable components in the reconstruction process.  Our geometry cloak can work on various methods like LGM[1], as 3DGS requires explicit geometry features for 3D representation. By exploiting this vulnerability, we can effectively manipulate the reconstructed 3D results, which is a noteworthy issue given the popularity of 3DGS today. Our approach offers a fresh viewpoint on adversarial perturbations, preventing copyrighted images from being used by GS-based methods. This could potentially stimulate further research into improving the resilience of 3D reconstruction and addressing AI privacy concerns.

To further clarify our work, we have provided more experimental results on the rebuttal page.

**1. Extending to other methods**

Fig. R1 and Tab. R1 provide the qualitative and quantitative results when implementing our method on LGM [1].
Fig. R1 presents the reconstructed views and point cloud via LGM under different perturbating strategies. The reconstructed 3D model is undermined and manipulated via our geometry cloak.
In Tab. R1, we experiment on three default scenes in LGM and report the reconstructed results when applying Gaussian noise and our method to the input views.  The results show that our method can effectively disturb the quality of the reconstructed 3D model.

**2. Combining adv. tri-plane**

Fig. R2 (a) presents the visual results when combining perturbation on the tri-plane feature and geometry feature.  Combining the two does not improve the attack performance, as the tri-plane feature is a robust part of TGS that is difficult to disturb. Future work could focus on studying the components in the 3D reconstruction process that are vulnerable to disturbances.

**3. Perturbations with smaller/larger budget**

Fig. R2 (b-c) provide the visual results when employing a smaller/larger budget $\epsilon$. These two figures indicate that our method is insensitive to larger epsilon values, as high-intensity Gaussian noise struggles to disturb the geometric features of the reconstructed results.

**4. Tendency of performance degradation**

Fig. R3 (1-3) illustrate the quality of the reconstructed 3D results under different epsilon intensities. An obvious decrease in reconstruction quality occurs within the 0 to 4 intensity range.

**5. Convergence status under different budget**

Fig. R3 (4) presents the convergence status under different budgets.


**6. Quality of protected image**

Fig. R4 (1-3) shows the quality of the protected image under different perturbation strategies.

**7. Computational resources**

Fig. R4 (4) presents the time required to finish protection via our method.

**8. Viewing direction**

Fig. R5 (left) illustrates the visual results of embedding a watermark at different angles and observing it from different perspectives.

Tab. R1 (right) shows the metric results when embedding a watermark at different angles.

**9. Multi-character as watermarks**

Fig. R5 (right) presents the results when embedding multi-character as the side view.

**10. Robustness against image compression**

Tab. R1 (right) presents the results when the protected image is modified via common image operations.


[1] LGM: Large Multi-View Gaussian Model for High-Resolution 3D Content Creation. In ECCV 2024.

---

### Decision · Program_Chairs · 2024-09-25

**Decision:**

Accept (poster)

**Comment:**

The paper introduces Geometry Cloak, a method to protect copyrighted images from unauthorized 3D reconstruction using Triplane Gaussian Splatting (TGS). Reviewer kd52 appreciates the novel perturbation technique that degrades 3D geometry without affecting 2D image quality but notes its limited applicability beyond TGS. Reviewer ykxU commends the clear presentation and effectiveness, suggesting extensions to other models like LRM and LGM, and testing robustness under noise and compression. Reviewer inRn acknowledges the novelty but questions adaptability to other methods and potential impacts on legitimate image use. Reviewer rPf4 recognized the significance and thorough experiments but recommends adding quantitative metrics like PSNR. Reviewer u5Eb criticizes the limited scope, urging tests on additional reconstruction methods for broader practical effects.

The rebuttals clarifies the geometry cloak method extends beyond TGS to other 3D reconstruction methods like LGM, as these rely on explicit geometric features. They provide additional experiments demonstrating the method's effectiveness across various settings, addressing concerns about generalizability, robustness, computational resources, and impact on image quality. Overall the reviewers welcomed the changes except u5Eb, who stressed that the paper should be evaluated on the initial submission, not the rebuttal. However, other reviewers (and AC) all feel that the changes in the rebuttal has indeed broadened the scope of the paper and made the point valid.